# Deformable DETR: Deformable Transformers for End-to-End Object Detection

**Xizhou Zhu**[1*], **Weijie Su**[2*‡], **Lewei Lu**[1], **Bin Li**[2], **Xiaogang Wang**[1,3], **Jifeng Dai**[1†]
[1]SenseTime Research
[2]University of Science and Technology of China
[3]The Chinese University of Hong Kong
`{zhuwalter,luotto,daijifeng}@sensetime.com`
`jackroos@mail.ustc.edu.cn,binli@ustc.edu.cn`
`xgwang@ee.cuhk.edu.hk`

## Abstract

DETR has been recently proposed to eliminate the need for many hand-designed components in object detection while demonstrating good performance. However, it suffers from slow convergence and limited feature spatial resolution, due to the limitation of Transformer attention modules in processing image feature maps. To mitigate these issues, we proposed Deformable DETR, whose attention modules only attend to a small set of key sampling points around a reference. Deformable DETR can achieve better performance than DETR (especially on small objects) with $10\times$ less training epochs. Extensive experiments on the COCO benchmark demonstrate the effectiveness of our approach. Code is released at `https://github.com/fundamentalvision/Deformable-DETR`.

## 1 Introduction

Modern object detectors employ many hand-crafted components (Liu et al., 2020), e.g., anchor generation, rule-based training target assignment, non-maximum suppression (NMS) post-processing. They are not fully end-to-end. Recently, Carion et al. (2020) proposed DETR to eliminate the need for such hand-crafted components, and built the first fully end-to-end object detector, achieving very competitive performance. DETR utilizes a simple architecture, by combining convolutional neural networks (CNNs) and Transformer (Vaswani et al., 2017) encoder-decoders. They exploit the versatile and powerful relation modeling capability of Transformers to replace the hand-crafted rules, under properly designed training signals.

Despite its interesting design and good performance, DETR has its own issues: (1) It requires much longer training epochs to converge than the existing object detectors. For example, on the COCO (Lin et al., 2014) benchmark, DETR needs 500 epochs to converge, which is around 10 to 20 times slower than Faster R-CNN (Ren et al., 2015). (2) DETR delivers relatively low performance at detecting small objects. Modern object detectors usually exploit multi-scale features, where small objects are detected from high-resolution feature maps. Meanwhile, high-resolution feature maps lead to unacceptable complexities for DETR. The above-mentioned issues can be mainly attributed to the deficit of Transformer components in processing image feature maps. At initialization, the attention modules cast nearly uniform attention weights to all the pixels in the feature maps. Long training epoches is necessary for the attention weights to be learned to focus on sparse meaningful locations. On the other hand, the attention weights computation in Transformer encoder is of quadratic computation w.r.t. pixel numbers. Thus, it is of very high computational and memory complexities to process high-resolution feature maps.

In the image domain, deformable convolution (Dai et al., 2017) is of a powerful and efficient mechanism to attend to sparse spatial locations. It naturally avoids the above-mentioned issues. While it lacks the element relation modeling mechanism, which is the key for the success of DETR.

---

*Equal contribution. †Corresponding author. ‡Work is done during an internship at SenseTime Research.

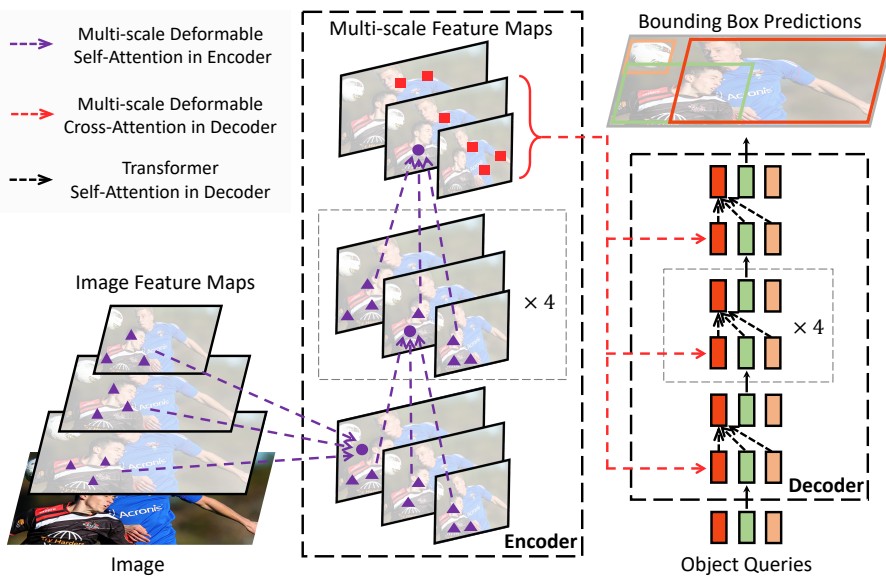

Figure 1: Illustration of the proposed Deformable DETR object detector.

In this paper, we propose *Deformable DETR*, which mitigates the slow convergence and high complexity issues of DETR. It combines the best of the sparse spatial sampling of deformable convolution, and the relation modeling capability of Transformers. We propose the *deformable attention module*, which attends to a small set of sampling locations as a pre-filter for prominent key elements out of all the feature map pixels. The module can be naturally extended to aggregating multi-scale features, without the help of FPN (Lin et al., 2017a). In Deformable DETR , we utilize (multi-scale) deformable attention modules to replace the Transformer attention modules processing feature maps, as shown in Fig. 1.

Deformable DETR opens up possibilities for us to exploit variants of end-to-end object detectors, thanks to its fast convergence, and computational and memory efficiency. We explore a simple and effective *iterative bounding box refinement* mechanism to improve the detection performance. We also try a *two-stage Deformable DETR*, where the region proposals are also generated by a vaiant of Deformable DETR, which are further fed into the decoder for iterative bounding box refinement.

Extensive experiments on the COCO (Lin et al., 2014) benchmark demonstrate the effectiveness of our approach. Compared with DETR, Deformable DETR can achieve better performance (especially on small objects) with $10\times$ less training epochs. The proposed variant of two-stage Deformable DETR can further improve the performance. Code is released at `https://github.com/fundamentalvision/Deformable-DETR`.

## 2    RELATED WORK

**Efficient Attention Mechanism.**    Transformers (Vaswani et al., 2017) involve both self-attention and cross-attention mechanisms. One of the most well-known concern of Transformers is the high time and memory complexity at vast key element numbers, which hinders model scalability in many cases. Recently, many efforts have been made to address this problem (Tay et al., 2020b), which can be roughly divided into three categories in practice.

The first category is to use pre-defined sparse attention patterns on keys. The most straightforward paradigm is restricting the attention pattern to be fixed local windows. Most works (Liu et al., 2018a; Parmar et al., 2018; Child et al., 2019; Huang et al., 2019; Ho et al., 2019; Wang et al., 2020a; Hu et al., 2019; Ramachandran et al., 2019; Qiu et al., 2019; Beltagy et al., 2020; Ainslie et al., 2020; Zaheer et al., 2020) follow this paradigm. Although restricting the attention pattern to a local neighborhood can decrease the complexity, it loses global information. To compensate, Child et al. (2019); Huang et al. (2019); Ho et al. (2019); Wang et al. (2020a) attend key elements

at fixed intervals to significantly increase the receptive field on keys. Beltagy et al. (2020); Ainslie et al. (2020); Zaheer et al. (2020) allow a small number of special tokens having access to all key elements. Zaheer et al. (2020); Qiu et al. (2019) also add some pre-fixed sparse attention patterns to attend distant key elements directly.

The second category is to learn data-dependent sparse attention. Kitaev et al. (2020) proposes a locality sensitive hashing (LSH) based attention, which hashes both the query and key elements to different bins. A similar idea is proposed by Roy et al. (2020), where k-means finds out the most related keys. Tay et al. (2020a) learns block permutation for block-wise sparse attention.

The third category is to explore the low-rank property in self-attention. Wang et al. (2020b) reduces the number of key elements through a linear projection on the size dimension instead of the channel dimension. Katharopoulos et al. (2020); Choromanski et al. (2020) rewrite the calculation of self-attention through kernelization approximation.

In the image domain, the designs of efficient attention mechanism (e.g., Parmar et al. (2018); Child et al. (2019); Huang et al. (2019); Ho et al. (2019); Wang et al. (2020a); Hu et al. (2019); Ramachandran et al. (2019)) are still limited to the first category. Despite the theoretically reduced complexity, Ramachandran et al. (2019); Hu et al. (2019) admit such approaches are much slower in implementation than traditional convolution with the same FLOPs (at least $3\times$ slower), due to the intrinsic limitation in memory access patterns.

On the other hand, as discussed in Zhu et al. (2019a), there are variants of convolution, such as deformable convolution (Dai et al., 2017; Zhu et al., 2019b) and dynamic convolution (Wu et al., 2019), that also can be viewed as self-attention mechanisms. Especially, deformable convolution operates much more effectively and efficiently on image recognition than Transformer self-attention. Meanwhile, it lacks the element relation modeling mechanism.

Our proposed deformable attention module is inspired by deformable convolution, and belongs to the second category. It only focuses on a small fixed set of sampling points predicted from the feature of query elements. Different from Ramachandran et al. (2019); Hu et al. (2019), deformable attention is just slightly slower than the traditional convolution under the same FLOPs.

**Multi-scale Feature Representation for Object Detection.** One of the main difficulties in object detection is to effectively represent objects at vastly different scales. Modern object detectors usually exploit multi-scale features to accommodate this. As one of the pioneering works, FPN (Lin et al., 2017a) proposes a top-down path to combine multi-scale features. PANet (Liu et al., 2018b) further adds an bottom-up path on the top of FPN. Kong et al. (2018) combines features from all scales by a global attention operation. Zhao et al. (2019) proposes a U-shape module to fuse multi-scale features. Recently, NAS-FPN (Ghiasi et al., 2019) and Auto-FPN (Xu et al., 2019) are proposed to automatically design cross-scale connections via neural architecture search. Tan et al. (2020) proposes the BiFPN, which is a repeated simplified version of PANet. Our proposed multi-scale deformable attention module can naturally aggregate multi-scale feature maps via attention mechanism, without the help of these feature pyramid networks.

## 3 REVISITING TRANSFORMERS AND DETR

**Multi-Head Attention in Transformers.** Transformers (Vaswani et al., 2017) are of a network architecture based on attention mechanisms for machine translation. Given a query element (e.g., a target word in the output sentence) and a set of key elements (e.g., source words in the input sentence), the *multi-head attention module* adaptively aggregates the key contents according to the attention weights that measure the compatibility of query-key pairs. To allow the model focusing on contents from different representation subspaces and different positions, the outputs of different attention heads are linearly aggregated with learnable weights. Let $q \in \Omega_q$ indexes a query element with representation feature $\boldsymbol{z}_q \in \mathbb{R}^C$, and $k \in \Omega_k$ indexes a key element with representation feature $\boldsymbol{x}_k \in \mathbb{R}^C$, where $C$ is the feature dimension, $\Omega_q$ and $\Omega_k$ specify the set of query and key elements, respectively. Then the multi-head attention feature is calculated by

$$\text{MultiHeadAttn}(\boldsymbol{z}_q, \boldsymbol{x}) = \sum_{m=1}^{M} \boldsymbol{W}_m \Big[ \sum_{k \in \Omega_k} A_{mqk} \cdot \boldsymbol{W}'_m \boldsymbol{x}_k \Big], \tag{1}$$

where $m$ indexes the attention head, $\boldsymbol{W}'_m \in \mathbb{R}^{C_v \times C}$ and $\boldsymbol{W}_m \in \mathbb{R}^{C \times C_v}$ are of learnable weights ($C_v = C/M$ by default). The attention weights $A_{mqk} \propto \exp\{\frac{\boldsymbol{z}_q^T \boldsymbol{U}_m^T \boldsymbol{V}_m \boldsymbol{x}_k}{\sqrt{C_v}}\}$ are normalized as $\sum_{k \in \Omega_k} A_{mqk} = 1$, in which $\boldsymbol{U}_m, \boldsymbol{V}_m \in \mathbb{R}^{C_v \times C}$ are also learnable weights. To disambiguate different spatial positions, the representation features $\boldsymbol{z}_q$ and $\boldsymbol{x}_k$ are usually of the concatenation/summation of element contents and positional embeddings.

There are two known issues with Transformers. One is Transformers need long training schedules before convergence. Suppose the number of query and key elements are of $N_q$ and $N_k$, respectively. Typically, with proper parameter initialization, $\boldsymbol{U}_m \boldsymbol{z}_q$ and $\boldsymbol{V}_m \boldsymbol{x}_k$ follow distribution with mean of 0 and variance of 1, which makes attention weights $A_{mqk} \approx \frac{1}{N_k}$, when $N_k$ is large. It will lead to ambiguous gradients for input features. Thus, long training schedules are required so that the attention weights can focus on specific keys. In the image domain, where the key elements are usually of image pixels, $N_k$ can be very large and the convergence is tedious.

On the other hand, the computational and memory complexity for multi-head attention can be very high with numerous query and key elements. The computational complexity of Eq. 1 is of $O(N_q C^2 + N_k C^2 + N_q N_k C)$. In the image domain, where the query and key elements are both of pixels, $N_q = N_k \gg C$, the complexity is dominated by the third term, as $O(N_q N_k C)$. Thus, the multi-head attention module suffers from a quadratic complexity growth with the feature map size.

**DETR.** DETR (Carion et al., 2020) is built upon the Transformer encoder-decoder architecture, combined with a set-based Hungarian loss that forces unique predictions for each ground-truth bounding box via bipartite matching. We briefly review the network architecture as follows.

Given the input feature maps $\boldsymbol{x} \in \mathbb{R}^{C \times H \times W}$ extracted by a CNN backbone (e.g., ResNet (He et al., 2016)), DETR exploits a standard Transformer encoder-decoder architecture to transform the input feature maps to be features of a set of object queries. A 3-layer feed-forward neural network (FFN) and a linear projection are added on top of the object query features (produced by the decoder) as the detection head. The FFN acts as the regression branch to predict the bounding box coordinates $\boldsymbol{b} \in [0,1]^4$, where $\boldsymbol{b} = \{b_x, b_y, b_w, b_h\}$ encodes the normalized box center coordinates, box height and width (relative to the image size). The linear projection acts as the classification branch to produce the classification results.

For the Transformer encoder in DETR, both query and key elements are of pixels in the feature maps. The inputs are of ResNet feature maps (with encoded positional embeddings). Let $H$ and $W$ denote the feature map height and width, respectively. The computational complexity of self-attention is of $O(H^2 W^2 C)$, which grows quadratically with the spatial size.

For the Transformer decoder in DETR, the input includes both feature maps from the encoder, and $N$ object queries represented by learnable positional embeddings (e.g., $N = 100$). There are two types of attention modules in the decoder, namely, cross-attention and self-attention modules. In the cross-attention modules, object queries extract features from the feature maps. The query elements are of the object queries, and key elements are of the output feature maps from the encoder. In it, $N_q = N$, $N_k = H \times W$ and the complexity of the cross-attention is of $O(HWC^2 + NHWC)$. The complexity grows linearly with the spatial size of feature maps. In the self-attention modules, object queries interact with each other, so as to capture their relations. The query and key elements are both of the object queries. In it, $N_q = N_k = N$, and the complexity of the self-attention module is of $O(2NC^2 + N^2 C)$. The complexity is acceptable with moderate number of object queries.

DETR is an attractive design for object detection, which removes the need for many hand-designed components. However, it also has its own issues. These issues can be mainly attributed to the deficits of Transformer attention in handling image feature maps as key elements: (1) DETR has relatively low performance in detecting small objects. Modern object detectors use high-resolution feature maps to better detect small objects. However, high-resolution feature maps would lead to an unacceptable complexity for the self-attention module in the Transformer encoder of DETR, which has a quadratic complexity with the spatial size of input feature maps. (2) Compared with modern object detectors, DETR requires many more training epochs to converge. This is mainly because the attention modules processing image features are difficult to train. For example, at initialization, the cross-attention modules are almost of average attention on the whole feature maps. While, at the end of the training, the attention maps are learned to be very sparse, focusing only on the object

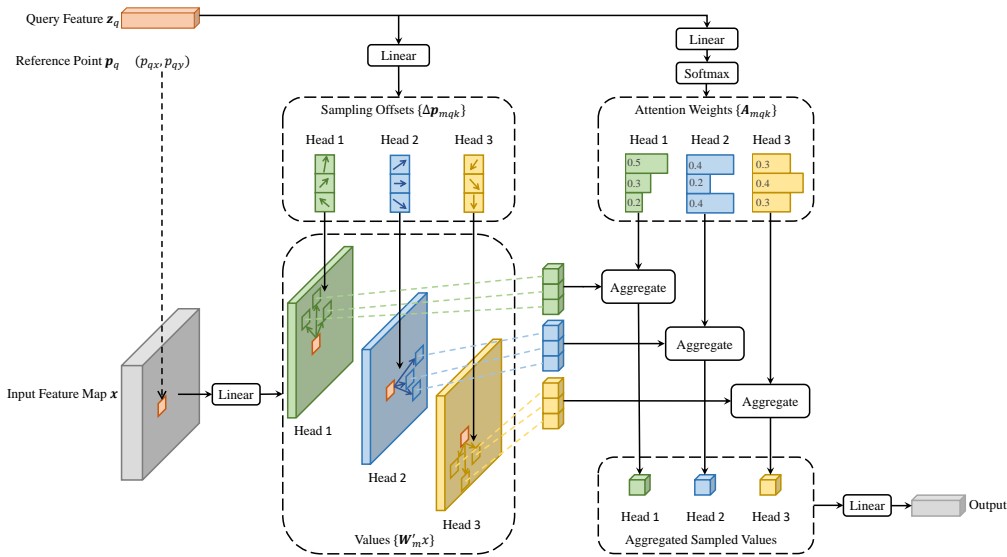

Figure 2: Illustration of the proposed deformable attention module.

extremities. It seems that DETR requires a long training schedule to learn such significant changes in the attention maps.

# 4 METHOD

## 4.1 DEFORMABLE TRANSFORMERS FOR END-TO-END OBJECT DETECTION

**Deformable Attention Module.** The core issue of applying Transformer attention on image feature maps is that it would look over all possible spatial locations. To address this, we present a *deformable attention module*. Inspired by deformable convolution (Dai et al., 2017; Zhu et al., 2019b), the deformable attention module only attends to a small set of key sampling points around a reference point, regardless of the spatial size of the feature maps, as shown in Fig. 2. By assigning only a small fixed number of keys for each query, the issues of convergence and feature spatial resolution can be mitigated.

Given an input feature map $\boldsymbol{x} \in \mathbb{R}^{C \times H \times W}$, let $q$ index a query element with content feature $\boldsymbol{z}_q$ and a 2-d reference point $\boldsymbol{p}_q$, the deformable attention feature is calculated by

$$\text{DeformAttn}(\boldsymbol{z}_q, \boldsymbol{p}_q, \boldsymbol{x}) = \sum_{m=1}^{M} \boldsymbol{W}_m \big[ \sum_{k=1}^{K} A_{mqk} \cdot \boldsymbol{W}'_m \boldsymbol{x}(\boldsymbol{p}_q + \Delta \boldsymbol{p}_{mqk}) \big], \tag{2}$$

where $m$ indexes the attention head, $k$ indexes the sampled keys, and $K$ is the total sampled key number ($K \ll HW$). $\Delta \boldsymbol{p}_{mqk}$ and $A_{mqk}$ denote the sampling offset and attention weight of the $k^{\text{th}}$ sampling point in the $m^{\text{th}}$ attention head, respectively. The scalar attention weight $A_{mqk}$ lies in the range $[0, 1]$, normalized by $\sum_{k=1}^{K} A_{mqk} = 1$. $\Delta \boldsymbol{p}_{mqk} \in \mathbb{R}^2$ are of 2-d real numbers with unconstrained range. As $\boldsymbol{p}_q + \Delta \boldsymbol{p}_{mqk}$ is fractional, bilinear interpolation is applied as in Dai et al. (2017) in computing $\boldsymbol{x}(\boldsymbol{p}_q + \Delta \boldsymbol{p}_{mqk})$. Both $\Delta \boldsymbol{p}_{mqk}$ and $A_{mqk}$ are obtained via linear projection over the query feature $\boldsymbol{z}_q$. In implementation, the query feature $\boldsymbol{z}_q$ is fed to a linear projection operator of $3MK$ channels, where the first $2MK$ channels encode the sampling offsets $\Delta \boldsymbol{p}_{mqk}$, and the remaining $MK$ channels are fed to a softmax operator to obtain the attention weights $A_{mqk}$.

The deformable attention module is designed for processing convolutional feature maps as key elements. Let $N_q$ be the number of query elements, when $MK$ is relatively small, the complexity of the deformable attention module is of $O(2N_q C^2 + \min(HWC^2, N_q KC^2))$ (See Appendix A.1 for details). When it is applied in DETR encoder, where $N_q = HW$, the complexity becomes $O(HWC^2)$, which is of linear complexity with the spatial size. When it is applied as the cross-attention modules

in DETR decoder, where $N_q = N$ ($N$ is the number of object queries), the complexity becomes $O(NKC^2)$, which is irrelevant to the spatial size $HW$.

**Multi-scale Deformable Attention Module.** Most modern object detection frameworks benefit from multi-scale feature maps (Liu et al., 2020). Our proposed deformable attention module can be naturally extended for multi-scale feature maps.

Let $\{x^l\}_{l=1}^L$ be the input multi-scale feature maps, where $x^l \in \mathbb{R}^{C \times H_l \times W_l}$. Let $\hat{p}_q \in [0, 1]^2$ be the normalized coordinates of the reference point for each query element $q$, then the multi-scale deformable attention module is applied as

$$\text{MSDeformAttn}(z_q, \hat{p}_q, \{x^l\}_{l=1}^L) = \sum_{m=1}^M W_m \big[ \sum_{l=1}^L \sum_{k=1}^K A_{mlqk} \cdot W_m' x^l(\phi_l(\hat{p}_q) + \Delta p_{mlqk}) \big], \quad (3)$$

where $m$ indexes the attention head, $l$ indexes the input feature level, and $k$ indexes the sampling point. $\Delta p_{mlqk}$ and $A_{mlqk}$ denote the sampling offset and attention weight of the $k^{\text{th}}$ sampling point in the $l^{\text{th}}$ feature level and the $m^{\text{th}}$ attention head, respectively. The scalar attention weight $A_{mlqk}$ is normalized by $\sum_{l=1}^L \sum_{k=1}^K A_{mlqk} = 1$. Here, we use normalized coordinates $\hat{p}_q \in [0, 1]^2$ for the clarity of scale formulation, in which the normalized coordinates $(0, 0)$ and $(1, 1)$ indicate the top-left and the bottom-right image corners, respectively. Function $\phi_l(\hat{p}_q)$ in Equation 3 re-scales the normalized coordinates $\hat{p}_q$ to the input feature map of the $l$-th level. The multi-scale deformable attention is very similar to the previous single-scale version, except that it samples $LK$ points from multi-scale feature maps instead of $K$ points from single-scale feature maps.

The proposed attention module will degenerate to deformable convolution (Dai et al., 2017), when $L = 1$, $K = 1$, and $W_m' \in \mathbb{R}^{C_v \times C}$ is fixed as an identity matrix. Deformable convolution is designed for single-scale inputs, focusing only on one sampling point for each attention head. However, our multi-scale deformable attention looks over multiple sampling points from multi-scale inputs. The proposed (multi-scale) deformable attention module can also be perceived as an efficient variant of Transformer attention, where a pre-filtering mechanism is introduced by the deformable sampling locations. When the sampling points traverse all possible locations, the proposed attention module is equivalent to Transformer attention.

**Deformable Transformer Encoder.** We replace the Transformer attention modules processing feature maps in DETR with the proposed multi-scale deformable attention module. Both the input and output of the encoder are of multi-scale feature maps with the same resolutions. In encoder, we extract multi-scale feature maps $\{x^l\}_{l=1}^{L-1}$ ($L = 4$) from the output feature maps of stages $C_3$ through $C_5$ in ResNet (He et al., 2016) (transformed by a $1 \times 1$ convolution), where $C_l$ is of resolution $2^l$ lower than the input image. The lowest resolution feature map $x^L$ is obtained via a $3 \times 3$ stride 2 convolution on the final $C_5$ stage, denoted as $C_6$. All the multi-scale feature maps are of $C = 256$ channels. Note that the top-down structure in FPN (Lin et al., 2017a) is not used, because our proposed multi-scale deformable attention in itself can exchange information among multi-scale feature maps. The constructing of multi-scale feature maps are also illustrated in Appendix A.2. Experiments in Section 5.2 show that adding FPN will not improve the performance.

In application of the multi-scale deformable attention module in encoder, the output are of multi-scale feature maps with the same resolutions as the input. Both the key and query elements are of pixels from the multi-scale feature maps. For each query pixel, the reference point is itself. To identify which feature level each query pixel lies in, we add a scale-level embedding, denoted as $e_l$, to the feature representation, in addition to the positional embedding. Different from the positional embedding with fixed encodings, the scale-level embedding $\{e_l\}_{l=1}^L$ are randomly initialized and jointly trained with the network.

**Deformable Transformer Decoder.** There are cross-attention and self-attention modules in the decoder. The query elements for both types of attention modules are of object queries. In the cross-attention modules, object queries extract features from the feature maps, where the key elements are of the output feature maps from the encoder. In the self-attention modules, object queries interact with each other, where the key elements are of the object queries. Since our proposed deformable attention module is designed for processing convolutional feature maps as key elements, we only replace each cross-attention module to be the multi-scale deformable attention module, while leaving the self-attention modules unchanged. For each object query, the 2-d normalized coordinate of the

reference point $\hat{p}_q$ is predicted from its object query embedding via a learnable linear projection followed by a sigmoid function.

Because the multi-scale deformable attention module extracts image features around the reference point, we let the detection head predict the bounding box as relative offsets w.r.t. the reference point to further reduce the optimization difficulty. The reference point is used as the initial guess of the box center. The detection head predicts the relative offsets w.r.t. the reference point. Check Appendix A.3 for the details. In this way, the learned decoder attention will have strong correlation with the predicted bounding boxes, which also accelerates the training convergence.

By replacing Transformer attention modules with deformable attention modules in DETR, we establish an efficient and fast converging detection system, dubbed as Deformable DETR (see Fig. 1).

## 4.2 Additional Improvements and Variants for Deformable DETR

Deformable DETR opens up possibilities for us to exploit various variants of end-to-end object detectors, thanks to its fast convergence, and computational and memory efficiency. Due to limited space, we only introduce the core ideas of these improvements and variants here. The implementation details are given in Appendix A.4.

**Iterative Bounding Box Refinement.** This is inspired by the iterative refinement developed in optical flow estimation (Teed & Deng, 2020). We establish a simple and effective iterative bounding box refinement mechanism to improve detection performance. Here, each decoder layer refines the bounding boxes based on the predictions from the previous layer.

**Two-Stage Deformable DETR.** In the original DETR, object queries in the decoder are irrelevant to the current image. Inspired by two-stage object detectors, we explore a variant of Deformable DETR for generating region proposals as the first stage. The generated region proposals will be fed into the decoder as object queries for further refinement, forming a two-stage Deformable DETR.

In the first stage, to achieve high-recall proposals, each pixel in the multi-scale feature maps would serve as an object query. However, directly setting object queries as pixels will bring unacceptable computational and memory cost for the self-attention modules in the decoder, whose complexity grows quadratically with the number of queries. To avoid this problem, we remove the decoder and form an encoder-only Deformable DETR for region proposal generation. In it, each pixel is assigned as an object query, which directly predicts a bounding box. Top scoring bounding boxes are picked as region proposals. No NMS is applied before feeding the region proposals to the second stage.

## 5 Experiment

**Dataset.** We conduct experiments on COCO 2017 dataset (Lin et al., 2014). Our models are trained on the train set, and evaluated on the val set and test-dev set.

**Implementation Details.** ImageNet (Deng et al., 2009) pre-trained ResNet-50 (He et al., 2016) is utilized as the backbone for ablations. Multi-scale feature maps are extracted without FPN (Lin et al., 2017a). $M = 8$ and $K = 4$ are set for deformable attentions by default. Parameters of the deformable Transformer encoder are shared among different feature levels. Other hyper-parameter setting and training strategy mainly follow DETR (Carion et al., 2020), except that Focal Loss (Lin et al., 2017b) with loss weight of 2 is used for bounding box classification, and the number of object queries is increased from 100 to 300. We also report the performance of DETR-DC5 with these modifications for a fair comparison, denoted as DETR-DC5$^+$. By default, models are trained for 50 epochs and the learning rate is decayed at the 40-th epoch by a factor of 0.1. Following DETR(Carion et al., 2020), we train our models using Adam optimizer (Kingma & Ba, 2015) with base learning rate of $2 \times 10^{-4}$, $\beta_1 = 0.9$, $\beta_2 = 0.999$, and weight decay of $10^{-4}$. Learning rates of the linear projections, used for predicting object query reference points and sampling offsets, are multiplied by a factor of 0.1. Run time is evaluated on NVIDIA Tesla V100 GPU.

## 5.1 Comparison with DETR

As shown in Table 1, compared with Faster R-CNN + FPN, DETR requires many more training epochs to converge, and delivers lower performance at detecting small objects. Compared with

DETR, Deformable DETR achieves better performance (especially on small objects) with $10\times$ less training epochs. Detailed convergence curves are shown in Fig. 3. With the aid of iterative bounding box refinement and two-stage paradigm, our method can further improve the detection accuracy.

Our proposed Deformable DETR has on par FLOPs with Faster R-CNN + FPN and DETR-DC5. But the runtime speed is much faster ($1.6\times$) than DETR-DC5, and is just 25% slower than Faster R-CNN + FPN. The speed issue of DETR-DC5 is mainly due to the large amount of memory access in Transformer attention. Our proposed deformable attention can mitigate this issue, at the cost of unordered memory access. Thus, it is still slightly slower than traditional convolution.

Table 1: Comparision of Deformable DETR with DETR on COCO 2017 val set. DETR-DC5$^+$ denotes DETR-DC5 with Focal Loss and 300 object queries.

| Method | Epochs | AP | $AP_{50}$ | $AP_{75}$ | $AP_S$ | $AP_M$ | $AP_L$ | params | FLOPs | Training GPU hours | Inference FPS |
|---|---|---|---|---|---|---|---|---|---|---|---|
| Faster R-CNN + FPN | 109 | 42.0 | 62.1 | 45.5 | 26.6 | 45.4 | 53.4 | 42M | 180G | 380 | 26 |
| DETR | 500 | 42.0 | 62.4 | 44.2 | 20.5 | 45.8 | 61.1 | 41M | 86G | 2000 | 28 |
| DETR-DC5 | 500 | 43.3 | 63.1 | 45.9 | 22.5 | 47.3 | 61.1 | 41M | 187G | 7000 | 12 |
| DETR-DC5 | 50 | 35.3 | 55.7 | 36.8 | 15.2 | 37.5 | 53.6 | 41M | 187G | 700 | 12 |
| DETR-DC5$^+$ | 50 | 36.2 | 57.0 | 37.4 | 16.3 | 39.2 | 53.9 | 41M | 187G | 700 | 12 |
| Deformable DETR | 50 | 43.8 | 62.6 | 47.7 | 26.4 | 47.1 | 58.0 | 40M | 173G | 325 | 19 |
| + iterative bounding box refinement | 50 | 45.4 | 64.7 | 49.0 | 26.8 | 48.3 | 61.7 | 40M | 173G | 325 | 19 |
| ++ two-stage Deformable DETR | 50 | 46.2 | 65.2 | 50.0 | 28.8 | 49.2 | 61.7 | 40M | 173G | 340 | 19 |

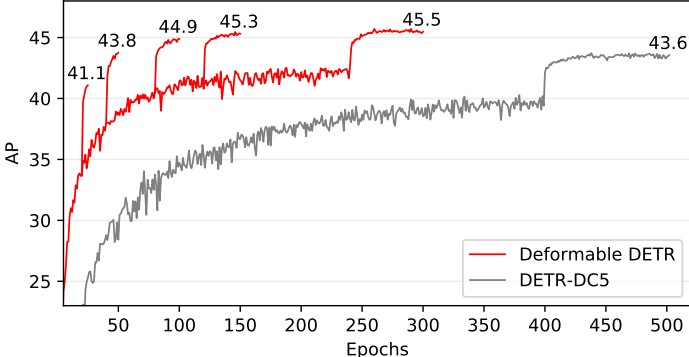

Figure 3: Convergence curves of Deformable DETR and DETR-DC5 on COCO 2017 val set. For Deformable DETR, we explore different training schedules by varying the epochs at which the learning rate is reduced (where the AP score leaps).

## 5.2 ABLATION STUDY ON DEFORMABLE ATTENTION

Table 2 presents ablations for various design choices of the proposed deformable attention module. Using multi-scale inputs instead of single-scale inputs can effectively improve detection accuracy with $1.7\%$ AP, especially on small objects with $2.9\%$ $AP_S$. Increasing the number of sampling points $K$ can further improve $0.9\%$ AP. Using multi-scale deformable attention, which allows information exchange among different scale levels, can bring additional $1.5\%$ improvement in AP. Because the cross-level feature exchange is already adopted, adding FPNs will not improve the performance. When multi-scale attention is not applied, and $K = 1$, our (multi-scale) deformable attention module degenerates to deformable convolution, delivering noticeable lower accuracy.

## 5.3 COMPARISON WITH STATE-OF-THE-ART METHODS

Table 3 compares the proposed method with other state-of-the-art methods. Iterative bounding box refinement and two-stage mechanism are both utilized by our models in Table 3. With ResNet-101 and ResNeXt-101 (Xie et al., 2017), our method achieves 48.7 AP and 49.0 AP without bells and whistles, respectively. By using ResNeXt-101 with DCN (Zhu et al., 2019b), the accuracy rises to 50.1 AP. With additional test-time augmentations, the proposed method achieves 52.3 AP.

Table 2: Ablations for deformable attention on COCO 2017 val set. "MS inputs" indicates using multi-scale inputs. "MS attention" indicates using multi-scale deformable attention. $K$ is the number of sampling points for each attention head on each feature level.

| MS inputs | MS attention | K | FPNs | AP | AP$_{50}$ | AP$_{75}$ | AP$_S$ | AP$_M$ | AP$_L$ |
|:---:|:---:|:---:|:---:|:---:|:---:|:---:|:---:|:---:|:---:|
| ✓ | ✓ | 4 | FPN (Lin et al., 2017a) | 43.8 | 62.6 | 47.8 | 26.5 | 47.3 | 58.1 |
| ✓ | ✓ | 4 | BiFPN (Tan et al., 2020) | 43.9 | 62.5 | 47.7 | 25.6 | 47.4 | 57.7 |
| | | 1 | | 39.7 | 60.1 | 42.4 | 21.2 | 44.3 | 56.0 |
| ✓ | | 1 | w/o | 41.4 | 60.9 | 44.9 | 24.1 | 44.6 | 56.1 |
| ✓ | | 4 | | 42.3 | 61.4 | 46.0 | 24.8 | 45.1 | 56.3 |
| ✓ | ✓ | 4 | | 43.8 | 62.6 | 47.7 | 26.4 | 47.1 | 58.0 |

Table 3: Comparison of Deformable DETR with state-of-the-art methods on COCO 2017 test-dev set. "TTA" indicates test-time augmentations including horizontal flip and multi-scale testing.

| Method | Backbone | TTA | AP | AP$_{50}$ | AP$_{75}$ | AP$_S$ | AP$_M$ | AP$_L$ |
|:---|:---:|:---:|:---:|:---:|:---:|:---:|:---:|:---:|
| FCOS (Tian et al., 2019) | ResNeXt-101 | | 44.7 | 64.1 | 48.4 | 27.6 | 47.5 | 55.6 |
| ATSS (Zhang et al., 2020) | ResNeXt-101 + DCN | ✓ | 50.7 | 68.9 | 56.3 | 33.2 | 52.9 | 62.4 |
| TSD (Song et al., 2020) | SENet154 + DCN | ✓ | 51.2 | 71.9 | 56.0 | 33.8 | 54.8 | 64.2 |
| EfficientDet-D7 (Tan et al., 2020) | EfficientNet-B6 | | 52.2 | 71.4 | 56.3 | - | - | - |
| Deformable DETR | ResNet-50 | | 46.9 | 66.4 | 50.8 | 27.7 | 49.7 | 59.9 |
| Deformable DETR | ResNet-101 | | 48.7 | 68.1 | 52.9 | 29.1 | 51.5 | 62.0 |
| Deformable DETR | ResNeXt-101 | | 49.0 | 68.5 | 53.2 | 29.7 | 51.7 | 62.8 |
| Deformable DETR | ResNeXt-101 + DCN | | 50.1 | 69.7 | 54.6 | 30.6 | 52.8 | 64.7 |
| Deformable DETR | ResNeXt-101 + DCN | ✓ | 52.3 | 71.9 | 58.1 | 34.4 | 54.4 | 65.6 |

## 6 CONCLUSION

Deformable DETR is an end-to-end object detector, which is efficient and fast-converging. It enables us to explore more interesting and practical variants of end-to-end object detectors. At the core of Deformable DETR are the (multi-scale) deformable attention modules, which is an efficient attention mechanism in processing image feature maps. We hope our work opens up new possibilities in exploring end-to-end object detection.

ACKNOWLEDGMENTS

The work is supported by the National Key R&D Program of China (2020AAA0105200), Beijing Academy of Artificial Intelligence, and the National Natural Science Foundation of China under grand No.U19B2044 and No.61836011.

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

## A  APPENDIX

### A.1  COMPLEXITY FOR DEFORMABLE ATTENTION

Supposes the number of query elements is $N_q$, in the deformable attention module (see Equation 2), the complexity for calculating the sampling coordinate offsets $\Delta \boldsymbol{p}_{mqk}$ and attention weights $A_{mqk}$ is of $O(3N_q CMK)$. Given the sampling coordinate offsets and attention weights, the complexity of computing Equation 2 is $O(N_q C^2 + N_q KC^2 + 5N_q KC)$, where the factor of 5 in $5N_q KC$ is because of bilinear interpolation and the weighted sum in attention. On the other hand, we can also calculate $\boldsymbol{W}'_m \boldsymbol{x}$ before sampling, as it is independent to query, and the complexity of computing Equation 2 will become as $O(N_q C^2 + HWC^2 + 5N_q KC)$. So the overall complexity of deformable attention is $O(N_q C^2 + \min(HWC^2, N_q KC^2) + 5N_q KC + 3N_q CMK)$. In our experiments, $M = 8$, $K \leq 4$ and $C = 256$ by default, thus $5K + 3MK < C$ and the complexity is of $O(2N_q C^2 + \min(HWC^2, N_q KC^2))$.

### A.2  CONSTRUCTING MULT-SCALE FEATURE MAPS FOR DEFORMABLE DETR

As discussed in Section 4.1 and illustrated in Figure 4, the input multi-scale feature maps of the encoder $\{\boldsymbol{x}^l\}_{l=1}^{L-1}$ ($L = 4$) are extracted from the output feature maps of stages $C_3$ through $C_5$ in ResNet (He et al., 2016) (transformed by a $1 \times 1$ convolution). The lowest resolution feature map $\boldsymbol{x}^L$ is obtained via a $3 \times 3$ stride 2 convolution on the final $C_5$ stage. Note that FPN (Lin et al., 2017a) is not used, because our proposed multi-scale deformable attention in itself can exchange information among multi-scale feature maps.

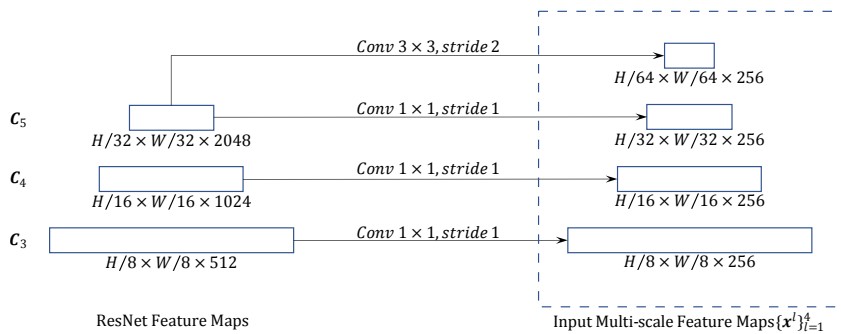

Figure 4: Constructing mult-scale feature maps for Deformable DETR.

### A.3  BOUNDING BOX PREDICTION IN DEFORMABLE DETR

Since the multi-scale deformable attention module extracts image features around the reference point, we design the detection head to predict the bounding box as relative offsets w.r.t. the reference point to further reduce the optimization difficulty. The reference point is used as the initial guess of the box center. The detection head predicts the relative offsets w.r.t. the reference point $\hat{\boldsymbol{p}}_q = (\hat{p}_{qx}, \hat{p}_{qy})$, i.e., $\hat{\boldsymbol{b}}_q = \{\sigma(b_{qx} + \sigma^{-1}(\hat{p}_{qx})), \sigma(b_{qy} + \sigma^{-1}(\hat{p}_{qy})), \sigma(b_{qw}), \sigma(b_{qh})\}$, where $b_{q\{x,y,w,h\}} \in \mathbb{R}$ are predicted by the detection head. $\sigma$ and $\sigma^{-1}$ denote the sigmoid and the inverse sigmoid function, respectively. The usage of $\sigma$ and $\sigma^{-1}$ is to ensure $\hat{\boldsymbol{b}}$ is of normalized coordinates, as $\hat{\boldsymbol{b}}_q \in [0, 1]^4$. In this way, the learned decoder attention will have strong correlation with the predicted bounding boxes, which also accelerates the training convergence.

### A.4  MORE IMPLEMENTATION DETAILS

**Iterative Bounding Box Refinement.**  Here, each decoder layer refines the bounding boxes based on the predictions from the previous layer. Suppose there are $D$ number of decoder layers (e.g., $D = 6$), given a normalized bounding box $\hat{\boldsymbol{b}}_q^{d-1}$ predicted by the $(d-1)$-th decoder layer, the $d$-th

decoder layer refines the box as

$$\hat{\boldsymbol{b}}_q^d = \{\sigma(\Delta b_{qx}^d + \sigma^{-1}(\hat{b}_{qx}^{d-1})), \sigma(\Delta b_{qy}^d + \sigma^{-1}(\hat{b}_{qy}^{d-1})), \sigma(\Delta b_{qw}^d + \sigma^{-1}(\hat{b}_{qw}^{d-1})), \sigma(\Delta b_{qh}^d + \sigma^{-1}(\hat{b}_{qh}^{d-1}))\},$$

where $d \in \{1, 2, ..., D\}$, $\Delta b_{q\{x,y,w,h\}}^d \in \mathbb{R}$ are predicted at the $d$-th decoder layer. Prediction heads for different decoder layers do not share parameters. The initial box is set as $\hat{b}_{qx}^0 = \hat{p}_{qx}$, $\hat{b}_{qy}^0 = \hat{p}_{qy}$, $\hat{b}_{qw}^0 = 0.1$, and $\hat{b}_{qh}^0 = 0.1$. The system is robust to the choice of $b_{qw}^0$ and $b_{qh}^0$. We tried setting them as 0.05, 0.1, 0.2, 0.5, and achieved similar performance. To stabilize training, similar to Teed & Deng (2020), the gradients only back propagate through $\Delta b_{q\{x,y,w,h\}}^d$, and are blocked at $\sigma^{-1}(\hat{b}_{q\{x,y,w,h\}}^{d-1})$.

In iterative bounding box refinement, for the $d$-th decoder layer, we sample key elements respective to the box $\hat{b}_q^{d-1}$ predicted from the $(d-1)$-th decoder layer. For Equation 3 in the cross-attention module of the $d$-th decoder layer, $(\hat{b}_{qx}^{d-1}, \hat{b}_{qy}^{d-1})$ serves as the new reference point. The sampling offset $\Delta \boldsymbol{p}_{mlqk}$ is also modulated by the box size, as $(\Delta p_{mlqkx} \hat{b}_{qw}^{d-1}, \Delta p_{mlqky} \hat{b}_{qh}^{d-1})$. Such modifications make the sampling locations related to the center and size of previously predicted boxes.

**Two-Stage Deformable DETR.** In the first stage, given the output feature maps of the encoder, a detection head is applied to each pixel. The detection head is of a 3-layer FFN for bounding box regression, and a linear projection for bounding box binary classification (i.e., foreground and background), respectively. Let $i$ index a pixel from feature level $l_i \in \{1, 2, ..., L\}$ with 2-d normalized coordinates $\hat{\boldsymbol{p}}_i = (\hat{p}_{ix}, \hat{p}_{iy}) \in [0, 1]^2$, its corresponding bounding box is predicted by

$$\hat{\boldsymbol{b}}_i = \{\sigma(\Delta b_{ix} + \sigma^{-1}(\hat{p}_{ix})), \sigma(\Delta b_{iy} + \sigma^{-1}(\hat{p}_{iy})), \sigma(\Delta b_{iw} + \sigma^{-1}(2^{l_i-1}s)), \sigma(\Delta b_{ih} + \sigma^{-1}(2^{l_i-1}s))\},$$

where the base object scale $s$ is set as 0.05, $\Delta b_{i\{x,y,w,h\}} \in \mathbb{R}$ are predicted by the bounding box regression branch. The Hungarian loss in DETR is used for training the detection head.

Given the predicted bounding boxes in the first stage, top scoring bounding boxes are picked as region proposals. In the second stage, these region proposals are fed into the decoder as initial boxes for the *iterative bounding box refinement*, where the positional embeddings of object queries are set as positional embeddings of region proposal coordinates.

**Initialization for Multi-scale Deformable Attention.** In our experiments, the number of attention heads is set as $M = 8$. In multi-scale deformable attention modules, $\boldsymbol{W}_m' \in \mathbb{R}^{C_v \times C}$ and $\boldsymbol{W}_m \in \mathbb{R}^{C \times C_v}$ are randomly initialized. Weight parameters of the linear projection for predicting $A_{mlqk}$ and $\Delta \boldsymbol{p}_{mlqk}$ are initialized to zero. Bias parameters of the linear projection are initialized to make $A_{mlqk} = \frac{1}{LK}$ and $\{\Delta \boldsymbol{p}_{1lqk} = (-k, -k), \Delta \boldsymbol{p}_{2lqk} = (-k, 0), \Delta \boldsymbol{p}_{3lqk} = (-k, k), \Delta \boldsymbol{p}_{4lqk} = (0, -k), \Delta \boldsymbol{p}_{5lqk} = (0, k), \Delta \boldsymbol{p}_{6lqk} = (k, -k), \Delta \boldsymbol{p}_{7lqk} = (k, 0), \Delta \boldsymbol{p}_{8lqk} = (k, k)\}$ ($k \in \{1, 2, ...K\}$) at initialization.

For *iterative bounding box refinement*, the initialized bias parameters for $\Delta \boldsymbol{p}_{mlqk}$ prediction in the decoder are further multiplied with $\frac{1}{2K}$, so that all the sampling points at initialization are within the corresponding bounding boxes predicted from the previous decoder layer.

### A.5 WHAT DEFORMABLE DETR LOOKS AT?

For studying what Deformable DETR looks at to give final detection result, we draw the gradient norm of each item in final prediction (i.e., x/y coordinate of object center, width/height of object bounding box, category score of this object) with respect to each pixel in the image, as shown in Fig. 5. According to Taylor's theorem, the gradient norm can reflect how much the output would be changed relative to the perturbation of the pixel, thus it could show us which pixels the model mainly relys on for predicting each item.

The visualization indicates that Deformable DETR looks at extreme points of the object to determine its bounding box, which is similar to the observation in DETR (Carion et al., 2020). More concretely, Deformable DETR attends to left/right boundary of the object for x coordinate and width, and top/bottom boundary for y coordinate and height. Meanwhile, different to DETR (Carion et al., 2020), our Deformable DETR also looks at pixels inside the object for predicting its category.

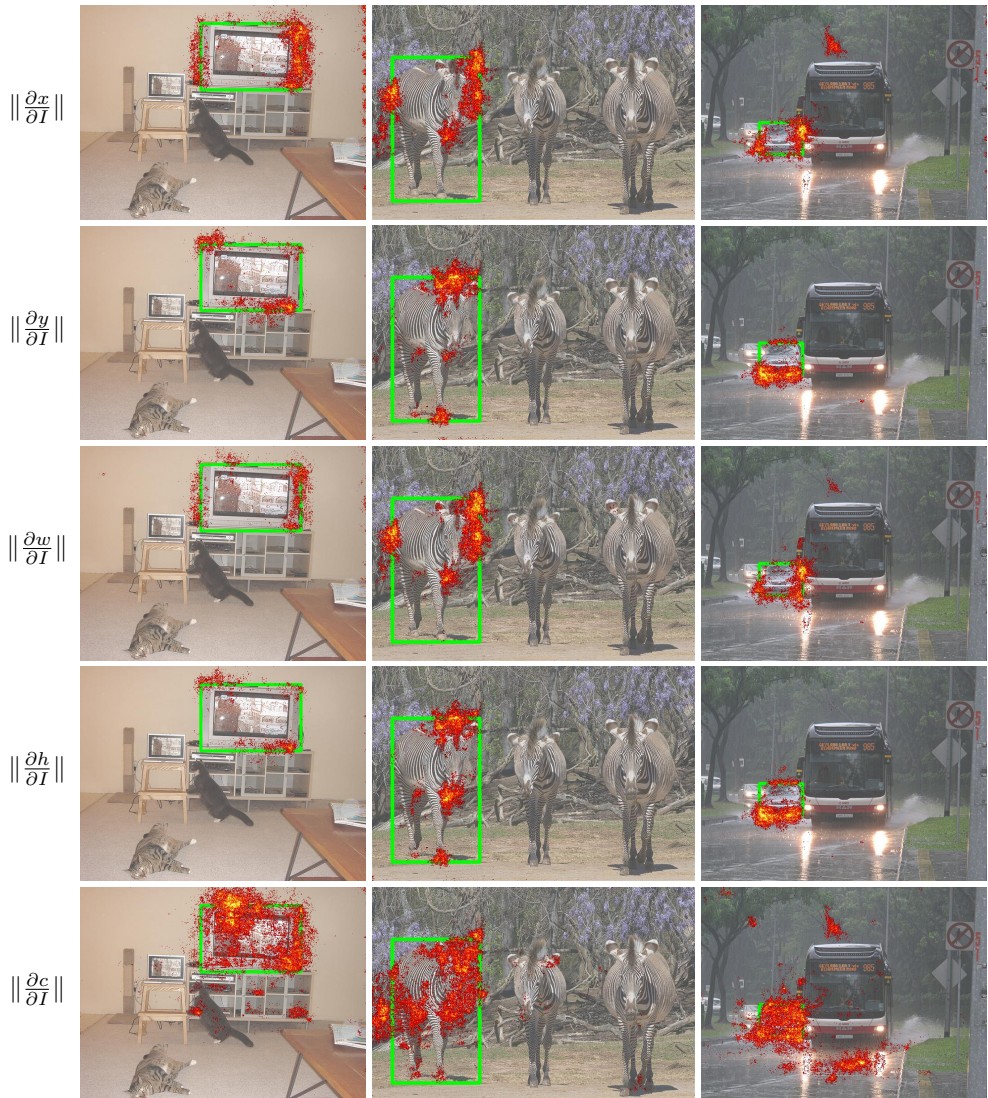

Figure 5: The gradient norm of each item (coordinate of object center $(x, y)$, width/height of object bounding box $w/h$, category score $c$ of this object) in final detection result with respect to each pixel in input image $I$.

## A.6 VISUALIZATION OF MULTI-SCALE DEFORMABLE ATTENTION

For better understanding learned multi-scale deformable attention modules, we visualize sampling points and attention weights of the last layer in encoder and decoder, as shown in Fig. 6. For readibility, we combine the sampling points and attention weights from feature maps of different resolutions into one picture.

Similar to DETR (Carion et al., 2020), the instances are already separated in the encoder of Deformable DETR. While in the decoder, our model is focused on the whole foreground instance instead of only extreme points as observed in DETR (Carion et al., 2020). Combined with the visualization of $\|\frac{\partial c}{\partial I}\|$ in Fig. 5, we can guess the reason is that our Deformable DETR needs not only extreme points but also interior points to detemine object category. The visualization also demonstrates that the proposed multi-scale deformable attention module can adapt its sampling points and attention weights according to different scales and shapes of the foreground object.

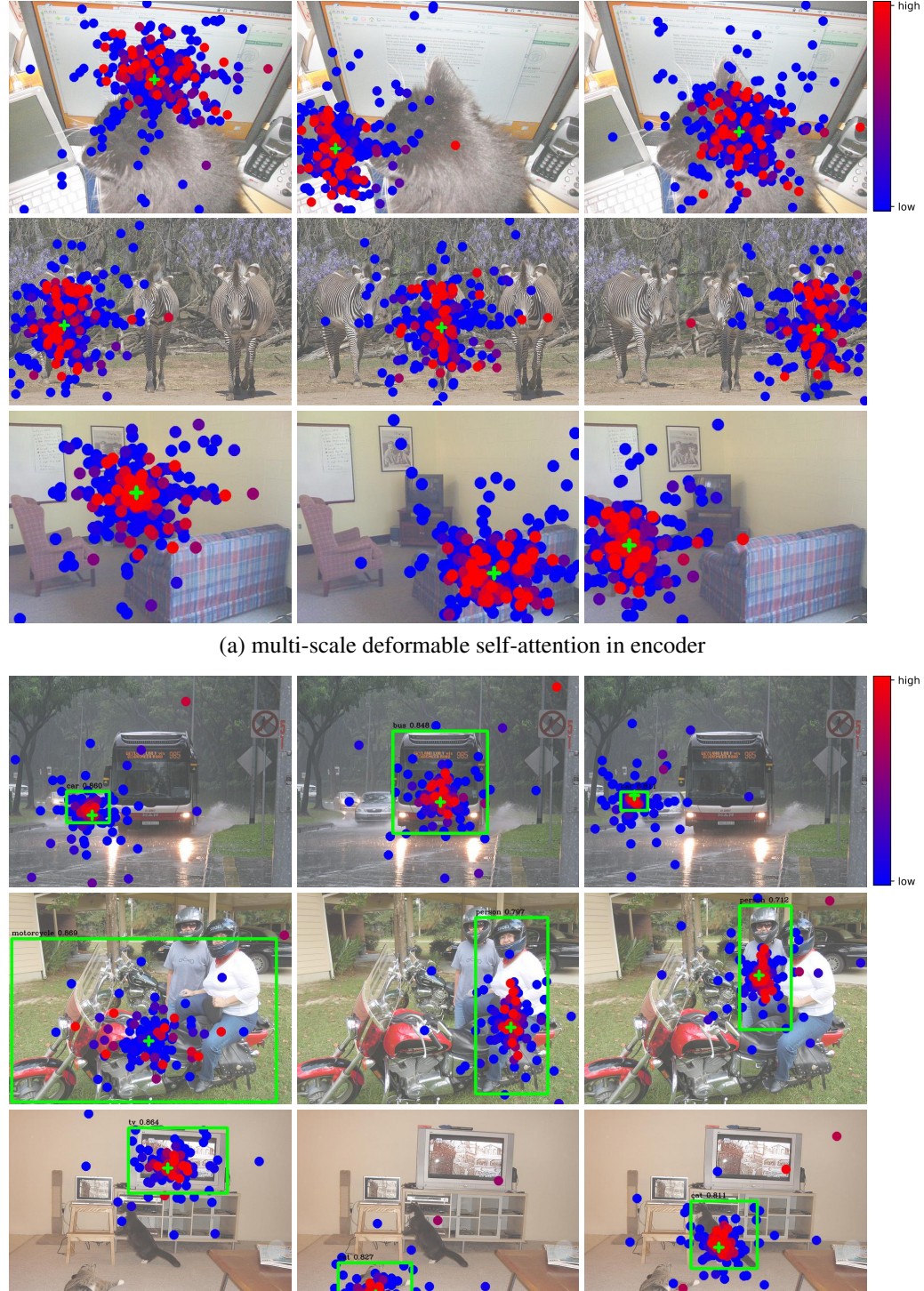

Figure 6: Visualization of multi-scale deformable attention. For readibility, we draw the sampling points and attention weights from feature maps of different resolutions in one picture. Each sampling point is marked as a filled circle whose color indicates its correspoinding attention weight. The reference point is shown as green cross marker, which is also equivalent to query point in encoder. In decoder, the predicted bounding box is shown as a green rectangle and the category and confidence score are texted just above it.

## A.7  NOTATIONS

Table 4: Lookup table for notations in the paper.

| Notation | Description |
|---|---|
| $m$ | index for attention head |
| $l$ | index for feature level of key element |
| $q$ | index for query element |
| $k$ | index for key element |
| $N_q$ | number of query elements |
| $N_k$ | number of key elements |
| $M$ | number of attention heads |
| $L$ | number of input feature levels |
| $K$ | number of sampled keys in each feature level for each attention head |
| $C$ | input feature dimension |
| $C_v$ | feature dimension at each attention head |
| $H$ | height of input feature map |
| $W$ | width of input feature map |
| $H^l$ | height of input feature map of $l^{th}$ feature level |
| $W^l$ | width of input feature map of $l^{th}$ feature level |
| $A_{mqk}$ | attention weight of $q^{th}$ query to $k^{th}$ key at $m^{th}$ head |
| $A_{mlqk}$ | attention weight of $q^{th}$ query to $k^{th}$ key in $l^{th}$ feature level at $m^{th}$ head |
| $\boldsymbol{z}_q$ | input feature of $q^{th}$ query |
| $\boldsymbol{p}_q$ | 2-d coordinate of reference point for $q^{th}$ query |
| $\hat{\boldsymbol{p}}_q$ | normalized 2-d coordinate of reference point for $q^{th}$ query |
| $\boldsymbol{x}$ | input feature map (input feature of key elements) |
| $\boldsymbol{x}_k$ | input feature of $k^{th}$ key |
| $\boldsymbol{x}^l$ | input feature map of $l^{th}$ feature level |
| $\Delta\boldsymbol{p}_{mqk}$ | sampling offset of $q^{th}$ query to $k^{th}$ key at $m^{th}$ head |
| $\Delta\boldsymbol{p}_{mlqk}$ | sampling offset of $q^{th}$ query to $k^{th}$ key in $l^{th}$ feature level at $m^{th}$ head |
| $\boldsymbol{W}_m$ | output projection matrix at $m^{th}$ head |
| $\boldsymbol{U}_m$ | input query projection matrix at $m^{th}$ head |
| $\boldsymbol{V}_m$ | input key projection matrix at $m^{th}$ head |
| $\boldsymbol{W}'_m$ | input value projection matrix at $m^{th}$ head |
| $\phi_l(\hat{\boldsymbol{p}})$ | unnormalized 2-d coordinate of $\hat{\boldsymbol{p}}$ in $l^{th}$ feature level |
| $\exp$ | exponential function |
| $\sigma$ | sigmoid function |
| $\sigma^{-1}$ | inverse sigmoid function |

