# OpenReview forum: "Deformable DETR: Deformable Transformers for End-to-End Object Detection"
_ICLR.cc/2021/Conference — ICLR 2021 Oral_

### Official Review · AnonReviewer2 · 2020-10-25
**Good paper; solid results on established benchmark; improved a recent detection model; some evaluations are missing;**

**Rating:** 7
**Confidence:** 4

**Review:**

### Summary
This paper aims to improve a very recent detection model -- DETR, which suffers from two issues: long training time and limited feature spatial resolution. Targeting these issues, this paper proposes (1) deformable attention (2) multi-scale processing (inputs/attention) for DETR, which have greatly reduced the  training time and improved the performance. Moreover, it develops two additional modules "iterative bounding box refinement" and  "two-stage framework", which help to achieve the SOTA detection results on COCO.

### Pros
 - This paper is insightful as it studies the two biggest issues of DETR. The vanilla attention is slow but hard to replace as the performance may be harmed. Similarly, it's not super straightforward to incorporate multi-scale processing into DETR due to the novel framework architecture. This paper has addressed these two issues, demonstrated outstanding performance on COCO, and properly studied the effectiveness of each single module.
 - The deformable attention module is a new kind of attention implementation. It samples a fix number of feature points on spatial feature map and thus greatly reduces the complexity.
 - The additional techniques make lots of sense to me, and improve the results greatly. I'm impressed that transformer based model can achieve very competitive  as shown in Tab.3.

### Cons
 - The deformable attention is considered the most important contribution of this paper and thus should be studied more thoroughly. Specifically, comparisons between this module and other "linear"/"efficient" implementations of attention should be performed. As discussed in related work, baselines like "pre-defined sparse attention", "data-dependent sparse attention", and "low-rank property attention" should be considered. This kind of comparisons will help us to understand better about the proposed module, and also to researchers from other field who are interested in using this.
  - Another missed baseline is DETR with multi-scale input and attention (vanilla version) for decently long epochs. I would want to know whether the proposed multi-scale thing could help the vanilla DETR.
 - Does the level embedding help? I don't find experiments to support this design choice.
 - If small objects are the issue, why not have a feature map of H/4 x W/4? Why are multi-scale feature maps constructed like in Appendix Fig.3?

### Minor
 - There are lots of notations. It's pretty hard for me to remember "mlqk, A, W, \phi, ...". I would recommend the authors providing a lookup table in Appendix.
 - Personally, I think the discussion about FPN in main paper is distractive. The authors may want to move all of them into Appendix.

### Questions
 - Do the last 5 rows in Tab.3 use "iterative bounding box refinement" and  "two-stage framework"? The difference between these 5 lines are just the backbone?
 - Fig.2 shows that Deformable DETR keeps improving over time. It doesn't converge at Ep.50. Why stick to this number in most experiments? How many epochs do the models in Tab.3 get trained?
 - In Tab.1, why do "iterative bounding box refinement" and "two-stage Deformable DETR" have no influence on FPS? Shouldn't the speed become slower as they are iterative and stage-wise. Does this FPS mean training FPS of Deformable DETR only? If that's the case, please also provide the total training time of everything.
- Please guide me to the definition/explanation of "DETR-DC5". Does it mean the backbone in DETE is changed to ResNet-DC5?

---

> ### Author Response · Authors · 2020-11-24
> **Reply to AnonReviewer2 (1/2)**
>
> We thank the reviewer for the careful reviews and constructive suggestions. We answer the questions as follows.
>
> ----------
> Q#1: Comparisons between the deformable attention module and other "linear"/"efficient" implementations of attention should be performed.
>
> A#1: Thanks for the great suggestion. It is very valuable to conduct a survey to benchmark these efficient attention mechanisms on object detection. However, most of these efficient attention mechanisms are developed for the NLP domain, which haven been rarely studied in the image domain. Thus, conducting this survey requires lots of effort, which is beyond the scope of this paper. On the other hand, as we discussed in the related works, in the image domain, the designs of efficient attention mechanism are still limited to using pre-defined sparse attention patterns. Despite the theoretically reduced complexity, such approaches are much slower in implementation than traditional convolution with the same FLOPs (at least 3× slower), due to the intrinsic limitation in memory access patterns. Compared to these efficient attention mechanisms in the image domain, our deformable attention module learns data-dependent sparse attention patterns, and just slightly slower than the traditional convolution under the same FLOPs.
>
> ----------
> Q#2: "Another missed baseline is DETR with multi-scale input and attention (vanilla version) for decently long epochs."
>
> A#2: Thanks for your suggestion. However, it is impossible to conduct such an experiment. In particular, training DETR with multi-scale inputs and vanilla transformer attention modules would require more than 500G of GPU memory (with batchsize of 1), which far exceeds the GPU memory limit. This is because the vanilla transformer attention modules suffer from a quadratic complexity w.r.t the number of input elements, which is very large in the case of multi-scale inputs.
>
> ----------
> Q#3: Effect of level embedding.
>
> A#3: Adding level embedding is a natural design to identify which feature level each query pixel lies in. It will not introduce any additional computations, but improves the AP with 0.5 points. Due to the page limit, we did not include this ablation experiment, which will be added in the revision.
>
> | Deformable DETR     | Epochs |  AP  | AP50 | AP75 | AP@S | AP@M | AP@L |
> | ------------------- | :----: | :--: | :--: | :--: | :--: | :--: | :--: |
> | w. level embedding  |   50   | 43.8 | 62.6 | 47.7 | 26.4 | 47.1 | 58.0 |
> | w/o level embedding |   50   | 43.3 | 62.3 | 47.2 | 25.9 | 46.5 | 57.6 |
>
> ----------
> Q#4: Why using multi-scale feature maps instead of a high-resolution feature map (e.g., H/4 x W/4) to mitigate the issue of small objects?
>
> A#4: It is a common practice in object detection, that large objects are detected on low-resolution feature maps while small objects are detected on high-resolution feature maps. As shown in [a], using multi-scale feature maps is more effective than using the single-scale high-resolution feature map in object detection.
>
> [a] Feature pyramid networks for object detection. In CVPR, 2017.
>
> ----------
> Minor:
>
> Q#5: Add a lookup table for notations in the Appendix. Move the discussion about FPN into Appendix.
>
> A#5: Thanks for the suggestions. We shall consider them in revision.

---

> > ### Author Response · Authors · 2020-11-24
> > **Reply to AnonReviewer2 (2/2)**
> >
> > Questions:
> >
> > ----------
> >
> > Q#6: Do the last 5 rows in Tab.3 use "iterative bounding box refinement" and "two-stage framework"? The difference between these 5 lines is just the backbone?
> >
> > A#6: Sorry for the confusion. The differences among the last 5 rows in Tab.3 are just the backbone and TTA. They all use "iterative bounding box refinement" and "two-stage framework". We shall clarify this in revision.
> >
> > ----------
> >
> > Q#7: Why stick to 50 epochs? How many epochs do the models in Tab.3 get trained?
> >
> > A#7: Models in Tab.3 are all trained for 50 epochs. The reason we use 50 epochs for most experiments is that it is within an affordable training time. Otherwise, the experiments would take too much time for us.
> >
> > ----------
> >
> > Q#8: In Tab.1, why do "iterative bounding box refinement" and "two-stage Deformable DETR" have no influence on FPS? Does this FPS mean training FPS of Deformable DETR only? If that's the case, please also provide the total training time of everything.
> >
> > A#8: "FPS" in Tab.1 refers to the inference speed as usual. To be precise, the inference speed of Deformable DETR is 19.4 FPS. Adding "iterative bounding box refinement" causes <0.1 FPS drop. Further adding "two-stage Deformable DETR" makes the inference speed drop to 18.8 FPS. The total training time of models in Tab.1 are shown as follows (measured on NVIDIA Tesla V100 GPU), which will be added in the revision:
> >
> > | Model                               | Epochs | Total Training Time (GPU hours) |
> > | ----------------------------------- | :----: | :-----------------------------: |
> > | Faster R-CNN + FPN                  |  109   |               380               |
> > | DETR                                |  500   |              2000               |
> > | DETR-DC5                            |  500   |              7000               |
> > | DETR-DC5                            |   50   |               700               |
> > | DETR-DC5+                           |   50   |               700               |
> > | Deformable DETR                     |   50   |               325               |
> > | + iterative bounding box refinement |   50   |               325               |
> > | ++ two-stage Deformable DETR        |   50   |               340               |
> >
> > ----------
> >
> > Q#9: Definition/explanation of "DETR-DC5".
> >
> > A#9: We quote the word of "DETR-DC5" from the original DETR paper (see Technical details in Page 8 of [b]). The backbone of DETR-DC5 is ResNet50 with dilated C5 stage.
> >
> > [b] End-to-end object detection with transformers. In ECCV, 2020.

---

### Official Review · AnonReviewer3 · 2020-10-26
**This paper introduces Deformable DETR: A modified version of the recent DETR paper for end-to-end object detection with transformers**

**Rating:** 8
**Confidence:** 4

**Review:**

The main contribution is a new attention module called deformable attention module. Like deformable convolution, it adds a translation term into the expression of the transformer, allowing a sparse spatial sampling. The resulting model is very interesting in terms of convergence and complexity compared to the original DETR. A Multi-scale deformable attention module is also proposed. it needs to add a scale function in the attention module equation. Experiments shows that it increases the AP detection rate on MSCOCO compared to FasterR-CNN and DETR.
Contributions are clearly state and validated. The complexity study is very interesting and shows the interest of deformable attention module.
Figure 1 presents a general view of the model. Since the deformable attention module is the core of the contribution, it should be interesting to add a figure dedicated to this component. Combined with eq.2, it will give a better understanding of the method.
It seems that the Axial-DeepLab paper presented in ECCV-2020 misses in the references. This paper proposes a simple strategy for attention modules that also reduce complexity.
Results clearly show that deformable DETR provides better AP than DETR for less training-epochs. Moreover, the convergence is better than for Faster R-CNN (FPN).
One of the concerns with deformable convolution is that the computation speed is slower than classical convolution. The same drawback appears with deformable attention modules. Fps decrease from 26 to 19 compared to DETR. It should be interesting to also report fps in the state of the art comparison table 3.

---

> ### Author Response · Authors · 2020-11-24
> **Reply to AnonReviewer3**
>
> We thank the reviewer for the careful reviews and constructive suggestions. We answer the questions as follows.
>
> ----------
> Q#1: "Since the deformable attention module is the core of the contribution, it should be interesting to add a figure dedicated to this component."
>
> A#1: Thanks for the good suggestion. Due to the page limit, we did not include the figure that demonstrates the deformable attention module. We shall add it in the revision, where one more page is available.
>
> ----------
> Q#2: "It seems that the Axial-DeepLab paper presented in ECCV-2020 misses in the references."
>
> A#2: Thanks for your kind reminder. We shall add it in the revision.
>
> ----------
> Q#3: The computation speed of deformable attention modules is slower than classical convolution.
>
> A#3: As we discussed in the Related Work and Section 5.1, the vanilla Transformer attention suffers from the large amount of memory access, meanwhile the previously proposed sparse attention mechanisms suffer from the intrinsic limitation in memory access patterns. Our proposed deformable attention can mitigate these issues, but at the cost of small amount of unordered memory access. Thus, it is still slightly slower than the traditional convolution under the same FLOPs.
>
> On the other hand, the speed issue is because modern GPUs are equipped with highly optimized hardware and implementations for traditional convolution, while the random memory access required by the deformable attention modules is hardware unfriendly. The improvement in hardware and implementations may mitigate this issue.
>
> ----------
> Q#4: "It should be interesting to also report fps in the state of the art comparison table 3."
>
> A#4: It is difficult to compare FPS fairly with other methods at system level in table 3, because they use different codebases, DL frameworks (TensorFlow, PyTorch, etc.), and hardware platforms. We would try our best to add this comparison in revision.

---

### Official Review · AnonReviewer4 · 2020-10-27
**Official Blind Review #4**

**Rating:** 8
**Confidence:** 5

**Review:**

Summary:

This paper proposes Deformable DETR with multi-scale deformable attention modules to solve the problems of DETR: slow convergence and limited feature spatial resolution.  In particular, it has faster convergence and achieves better performance(especially on small objects) than DETR.

Reasons for score:

Overall, I vote for accepting. I like this paper because it solves the main problems suffered by DETR. My major concern is about the clarity of the paper and some additional ablation studies (see cons below). Hopefully the authors can address my concern in the rebuttal period.

Pros:

1.This paper solves the main problems suffered by DETR: slow convergence and limited feature spatial resolution. In my opinion, it makes DETR more practical.

2.The proposed Deformable DETR can obtain multi-scale features without a huge cost. In this way, it can be optimized easily and detect objects precisely, especially small objects.

3.This paper also introduces some improvements and variants to boost the performance of Deformable DETR.

Cons:

1.In the experiments, focal loss is used for bounding box classification. What is the reason for this choice? In addition, the number of object queries is increased from 100 to 300. Why? In the test, how to choose 100 objects from 300 objects?

2.From Table 1, we can find DETR (500 epochs) has better performance than Deformable DETR on large objects (61.1 vs. 58.0), though the overall performance of Deformable DETR is better. Why?

3.In the Table 1, there is only DETR-DC5+ (50 epochs). Could you provide DETR-DC5+ (500 epochs) ?

Questions during rebuttal period:

Please address and clarify the cons above

---

> ### Author Response · Authors · 2020-11-24
> **Reply to AnonReviewer4**
>
> We thank the reviewer for the careful reviews and constructive suggestions. We answer the questions as follows.
>
> ----------
> Q#1(a): "In the experiments, focal loss is used for bounding box classification. What is the reason for this choice? In addition, the number of object queries is increased from 100 to 300. Why?
>
> A#1(a): These two modifications are both general improvements for object detectors, which are also applicable on vanilla DETR. We also include an improved version of DETR (namely "DETR-DC5+") in table 1 for a fair comparison.
>
> Q#1(b): "In the test, how to choose 100 objects from 300 objects?"
>
> A#1(b): For each image, following the standard evaluation protocol for COCO detection, the top-scored 100 detections (across all categories) will be chosen as the final predictions.
>
> ----------
> Q#2: "From Table 1, we can find DETR (500 epochs) has better performance than Deformable DETR on large objects (61.1 vs. 58.0), though the overall performance of Deformable DETR is better. Why?"
>
> A#2: Indeed, we also find it is interesting that the original DETR is better than our Deformable DETR on large objects. However, we still don’t know what the reason is. Further study is needed.
>
> ----------
> Q#3: "In Table 1, there is only DETR-DC5+ (50 epochs). Could you provide DETR-DC5+ (500 epochs)?"
>
> A#3: Thanks for your suggestion. However, training DETR-DC5 for 500 epochs is extremely slow. Around 18 days are required on 16 NVIDIA Tesla V100 GPUs. Due to limited computing resources, it is not affordable for us. By the way, from this point, Deformable DETR opens up the possibilities to explore these variants thanks to its fast convergence and efficiency.

---

### Official Review · AnonReviewer1 · 2020-10-30
**It solves the slow convergence problem in the recent successful DETR framework and obtains SOTA results**

**Rating:** 9
**Confidence:** 5

**Review:**

As a new framework for object detection, DETR is very important. However,  it suffers from slow convergence and limited feature spatial resolution. This paper proposes deformable attention, which attends to a small set of sampling locations rather than all the locations in the original DETR. Besides, the paper applies multi-scale deformable attention for better results.

The paper is well-written and obtains very impressive results. Traning for only 50 epochs, deformable Detr obtain results similar to DETR which is trained for 500 epochs. By implementing a two-stage detector based on deformable Detr, the paper obtain state-of-the-art object detection results with a very high AP (52.3) on AP.

 A few suggestions for improving the paper are given as follows.
 (1) The training and testing times could be reported in the paper, which is useful for other researchers to implement and use this method.
 (2) Some related methods on sparse connected self-attention/transformer [a,b,c] should be cited and discussed.

 [a] Representative Graph Neural Network, CVPR 2020
 [b] Dynamic Graph Message Passing Networks, CVPR 2020
 [c] CCNet: Criss-Cross Attention for Semantic Segmentation in ICCV 19 & TPAMI 2020

---

> ### Author Response · Authors · 2020-11-24
> **Reply to AnonReviewer1**
>
> We thank the reviewer for the careful reviews and constructive suggestions. We answer the questions as follows.
>
> ----------
> Q#1: "The training and testing times could be reported in the paper, which is useful for other researchers to implement and use this method."
>
> A#1: Thanks for your good suggestion. The testing time is already reported in the column "FPS" of Table 1. The total training time of each model in Table 1 is shown as follows (measured on NVIDIA Tesla V100 GPU), which will be added in the revision:
>
> | Model                               | Epochs | Total Training Time (GPU hours) |
> | ----------------------------------- | :----: | :-----------------------------: |
> | Faster R-CNN + FPN                  |  109   |               380               |
> | DETR                                |  500   |              2000               |
> | DETR-DC5                            |  500   |              7000               |
> | DETR-DC5                            |   50   |               700               |
> | DETR-DC5+                           |   50   |               700               |
> | Deformable DETR                     |   50   |               325               |
> | + iterative bounding box refinement |   50   |               325               |
> | ++ two-stage Deformable DETR        |   50   |               340               |
>
> ----------
> Q#2: "Some related methods on sparse connected self-attention/transformer [a,b,c] should be cited and discussed."
>
> A#2: Thanks for your kind reminder. We shall cite and discuss them in revision. By the way, [c] has already been cited.
>
> [a] Representative Graph Neural Network, CVPR 2020
>
> [b] Dynamic Graph Message Passing Networks, CVPR 2020
>
> [c] CCNet: Criss-Cross Attention for Semantic Segmentation in ICCV 19 & TPAMI 2020

---

### Public Comment · ~Nicolas_Carion1 · 2020-11-10
**Is deformable attention an attention mechanism?**

Hello,

I have a question about the "Deformable Attention Module" presented in this paper. According to the text following equation (2), the attention coefficients $A_{m,q,k}$ "are obtained via linear projection over the query feature". In my opinion, that does not constitute an attention mechanism per-se, since by definition an attention should incorporate both features from the query and the key (the most popular instantiation of an attention mechanism being dot-product attention).
In other words, the presented mechanism aggregates features in a neighborhood irrespective of what the actual features are.
From a purely terminological point of view, the role played by these $A_{mqk}$ is more akin to a gating, and in my opinion a better name for the proposed module would be something like "multi-resolution gated deformable convolution". What is your opinion on this?
As a side note, in the proposed formulation, the $A_{m,q,k}$ could very well be computed as a dot-product as well (between the query and each of the sampled point), making it a "true" attention mechanism. Have you tried such thing?

Aside from the numerical results, which are impressive, it would provide invaluable insights to showcase visualizations of the "attention" maps in the encoder and the decoder, similar to the original DETR paper. In particular, is the proposed model learning an approximation of what the original DETR attention attends to, or is it attending to completely different things? Do you still observe instance separation inside the encoder?

The two additions on top of the vanilla model (namely "two stage detr" and "iterative bounding box refinement") are interesting and seem to be directly applicable to the original DETR as well. Do you have a ball-park estimate of the performance reached when applying them on DETR-DC5?
As a side note, the gains obtained thanks to the iterative bounding box refinement might hint at limitations in the original auxiliary-loss formulation, or the L1+GIOU combinations. It would be interesting to analyze this more deeply. It's surprising that it helps mainly for large objects in the deformable setting, I would have guessed it would help for small instead.

Finally, an ablation that would be valuable in my opinion would be to more carefully disentangle the effect of the multi-resolution and the effect of the proposed deformable module. In particular, in table2, the only experiment with no multi-scale input is done with k=1. What would be the performance with k=4 or even k=8 (which would be the most similar to DETR)? I'm curious to see if the proposed deformable module can match original DETR's performance on a low-res feature map, and if it requires long training schedule to do so. That would help understanding where the convergence speed boost comes from, and the exact performance trade-offs that are being made by using this different deformable mechanism as opposed to traditional attention.

---

> ### Author Response · Authors · 2020-11-24
> **Reply to Nicolas Carion (1/2)**
>
> Hi Nicolas,
>
> Thanks for your good questions and constructive suggestions. First of all, we answer the question in your title. In our understanding, attention mechanisms are known by the property that it enables the neural networks to focus more on relevant elements of the input than on irrelevant parts. It is not defined by the formulation of how the attention weights are calculate. In this context, the proposed deformable attention is indeed an attention mechanism. Then, we answer the other questions as follows.
>
> ----------
> Q#1: "...attention weight are obtained via linear projection over the query feature...by definition an attention should incorporate both features from the query and the key...As a side note, in the proposed formulation, the $A_{mqk}$ could very well be computed as a dot-product as well (between the query and each of the sampled point), making it a 'true' attention mechanism. Have you tried such thing?"
>
> A#1: Yes, we tried using dot-product to obtain the attention weight in some early experiments, where $K=1$ and other design choices are very similar with the default setting of Deformable DETR. It achieves on par performance (AP difference <0.5%) compared with that of linear projection. However, we experimentally found that using dot-product results ~25% slower speed than that of linear projection. Therefore, we choose to obtain the attention weight by linear projection for efficiency.
>
> We guess the reason of their very close performance is that the stacked convolutions and attention layers have provided enough contextual information for the query feature to determine the attention weights. We are also inspired by [a], which shows that the dot-product between the query and key content features (without the positional encodings) plays a minor role in the transformer self-attention.
>
> In terms of speed, using dot-product has the same computational complexity as the linear projection. The inefficiency of the dot-product may be related to the implementation. In comparison to linear projection, the dot-product requires additional random memory access for sampling key features and the batch matrix-matrix product, which may be the reason of inefficiency. In particular, for the dot-product, we first sample the key features (with the shape of $N_Q \times N_K \times C$) related to each query, and then apply the batch matrix-matrix product with the query features (with the shape of $N_Q \times C \times 1$), in order to obtain the attention weights (with the shape of $N_Q \times N_K$). Meanwhile, for the linear projection, we only need to compute a matrix multiplication between the query features (with the shape of $N_Q \times C$) and the weights of linear projection (with the shape of $C \times N_K$).
>
> We shall add these comparisons and discussions in the revision. Thanks for your suggestion.
>
> [a] An Empirical Study of Spatial Attention Mechanisms in Deep Networks. In ICCV, 2019.
>
> ----------
> Q#2: "showcase visualizations of the 'attention' maps in the encoder and the decoder"
>
> A#2: In the encoder, we observed very similar attention patterns between Deformable DETR and vanilla DETR, namely "instance separation". However, in the decoder, Deformable DETR focuses on the whole foreground instance instead of extreme points in vanilla DETR. We shall add the visualization in the revision.
>
> ----------
> Q#3: Adding "two-stage DETR" and "iterative bounding box refinement" to vanilla DETR.
>
> A#3: Actually, applying these two techniques to vanilla DETR is not straight-forward. Both techniques require a mechanism to guide the attention module w.r.t. the previously predicted bounding boxes, which is missing in vanilla DETR. Besides, trial and error are required to make these techniques compatible with vanilla DETR. However, vanilla DETR requires a very long training time. Due to limited computing resources, it is not affordable for us (also for many labs) to do many trials. From this point, Deformable DETR opens up the possibilities to explore these variants with its fast convergence and efficiency.
>
> ----------
> Q#4: "It's surprising that it helps mainly for large objects in the deformable setting, I would have guessed it would help for small instead."
>
> A#4: We are also surprised by the phenomenon that "iterative bounding box refinement" helps mainly for large objects, while "two-stage Deformable DETR" helps mainly for small objects. It is very interesting. However, we still don’t know what the reason is. Further study is needed.

---

> > ### Author Response · Authors · 2020-11-24
> > **Reply to Nicolas Carion (2/2)**
> >
> > ----------
> > Q#5: "disentangle the effect of the multi-resolution and the effect of the proposed deformable module"
> >
> > A#5: Thanks for your good suggestion. We tried training Deformable DETR (K = 4) with single-scale input feature maps (of stride 32) for 50 epochs and 150 epochs. The results are shown as follows. The single-scale model of 50 epochs is still slightly worse than DETR (500 epochs) in total AP, while the single-scale model of 150 epochs achieves on par accuracy with DETR (500 epochs). However, the phenomenon is still slightly different. The single-scale Deformable DETR performs better on small objects and performs worse on large objects, compared with vanilla DETR. Further study is needed.
> >
> > | Method                         |  K   | Epochs |  AP  | AP50 | AP75 | AP@S | AP@M | AP@L |
> > | ------------------------------ | :--: | :----: | :--: | :--: | :--: | :--: | :--: | :--: |
> > | DETR                           |  -   |  500   | 42.0 | 62.4 | 44.2 | 20.5 | 45.8 | 61.1 |
> > | DETR                           |  -   |   50   | 33.3 | 54.1 | 34.2 | 13.3 | 35.9 | 52.0 |
> > | Deformable DETR (single scale) |  4   |   50   | 40.5 | 60.6 | 43.0 | 21.5 | 44.7 | 56.7 |
> > | Deformable DETR (multi scale)  |  4   |   50   | 43.8 | 62.6 | 47.7 | 26.4 | 47.1 | 58.0 |
> > | DETR                           |  -   |  150   | 39.5 | 60.3 | 41.4 | 17.5 | 43.0 | 59.1 |
> > | Deformable DETR (single scale) |  4   |  150   | 41.6 | 61.9 | 44.6 | 22.8 | 45.3 | 58.0 |
> > | Deformable DETR (multi scale)  |  4   |  150   | 45.3 | 64.3 | 49.1 | 27.1 | 48.4 | 60.0 |

---

### Decision · Program_Chairs · 2021-01-07
**Final Decision**

**Decision:**

Accept (Oral)

**Comment:**

Accept. The paper proposes Deformable DETR that builds on DETR and solves the slow convergence and limited spatial resolution problem while getting impressive results.
The authors should think about comparing with other linear attention mechanisms to show the applicability of the method.